

# A description and evaluation of an air quality model nested within global and regional composition-climate models using MetUM

Lucy S. Neal[1], Mohit Dalvi[2], Gerd Folberth[2], Rachel N. McInnes[2,3], Paul Agnew[1], Fiona M. O'Connor[2], Nicholas H. Savage[1], and Marie Tilbee[1]

[1]Met Office, FitzRoy Road, Exeter, EX1 3PB, United Kingdom
[2]Met Office Hadley Centre, FitzRoy Road, Exeter, EX1 3PB, United Kingdom
[3]European Centre for Environment and Human Health, University of Exeter Medical School, Knowledge Spa, Royal Cornwall Hospital, Truro, TR1 3HD, UK

*Correspondence to:* L. S. Neal (lucy.neal@metoffice.gov.uk)

**Abstract.** There is a clear need for the development of modelling frameworks for both climate change and air quality to help inform policies for addressing these issues. This paper presents an initial attempt to develop a single modelling framework, by introducing a greater degree of consistency in the modelling framework by using a two-step, one-way nested configuration of models, from a

global composition-climate model (GCCM) (140 km resolution) to a regional composition-climate model covering Europe (RCCM) (50 km resolution) and finally to a high (12 km) resolution model over the UK (AQUM). The latter model is used to produce routine air quality forecasts for the UK. All three models are based on the Met Office's Unified Model (MetUM). In order to better understand the impact of resolution on the downscaling of projections of future climate and air quality, we

have used this nest of models to simulate a five year period using present-day emissions and under present-day climate conditions. We also consider the impact of running the higher resolution model with higher spatial resolution emissions, rather than simply regridding emissions from the RCCM. We present an evaluation of the models compared to in situ air quality observations over the UK, plus a comparison against an independent 1 km resolution gridded dataset, derived from a combination

of modelling and observations. We show that using a high resolution model over the UK has some benefits in improving air quality modelling, but that the use of higher spatial resolution emissions is important to capture local variations in concentrations, particularly for primary pollutants such as nitrogen dioxide and sulphur dioxide. For secondary pollutants such as ozone and the secondary component of PM10, the benefits of a higher resolution nested model are more limited and reasons

for this are discussed. This study confirms that the resolution of models is not the only factor in determining model performance - consistency between nested models is also important.



## 1   Introduction

Models for studying historical climate change and for projecting future climate have increased in
complexity and sophistication in recent years and the importance of including atmospheric compo-
sition as a component of such models is now well established (e.g. Eyring et al., 2013). Gas-phase
constituents, such as tropospheric ozone ($O_3$), exert a positive radiative forcing on climate (Steven-
son et al., 2013; Myhre et al., 2013) while the radiative forcings associated with aerosol-radiation
and aerosol-cloud interactions are partly masking the strong positive forcing associated with long-
lived greenhouse gases (Myhre et al., 2013). A changing climate, in turn, has an impact on both
natural emissions (e.g. Sanderson et al., 2003; Forkel and Knoche, 2006) and chemistry and aerosol
processes themselves (e.g. Jacob and Winner, 2009; Fiore et al., 2012; Allen et al., 2016), influenc-
ing atmospheric composition. Atmospheric composition and near-surface air quality are intricately
linked and poor air quality has impacts on human health (e.g. WHO, 2013b). In addition, surface $O_3$
can adversely impact crop growth (Sitch et al., 2007) while aerosols can potentially promote global
plant productivity by increasing the diffuse fraction of photosynthetically active radiation (Mercado
et al., 2009).

Given the interactions between atmospheric composition, air quality, and climate, it is essen-
tial that the development of climate change mitigation policies and air quality abatement strategies
are developed jointly and consider the full spectrum of co-benefits and trade-offs (e.g. vonSchnei-
demesser and Monks, 2013). As a result, there is a strong need for models that can simulate both
climate and air quality. Likewise, it is also necessary to develop modelling frameworks which can
dynamically downscale global climate and air quality projections to the regional scale, on which
population centres and crop locations vary significantly. Downscaling allows a greater level of de-
tail to be made explicit and analysed. Air pollutant concentrations exhibit a higher degree of spatial
inhomogeneity compared to typical meteorological fields and more highly resolved regional mod-
elling can improve the representation and evolution (due to more highly-resolved emissions and the
dependence of reaction rates on concentrations) of reactive species. A further imperative for higher
resolution modelling concerns the sensitivity of composition projections to the difference in meteo-
rology. For example, Kunkel et al. (2008) discuss the sensitivity of $O_3$ under regional climate change
to cumulus cloud parametrisations. In their review article, Jacob and Winner (2009) cite a number
of other examples where significantly differing model predictions are attributed to differences in air
pollution meteorology between global and higher resolution regional models.

Various modelling configurations have been employed in studies of regional air quality in the
context of present-day climate and under future climate change scenarios. A common approach has
been to use a global-regional climate model nest to provide meteorology and then use the stored
fields to drive an off-line chemistry transport model (CTM) (e.g. Lauwaet et al., 2013; Likhvar
et al., 2015). This approach was used, for example, to investigate the impacts of emission changes
on UK $O_3$ and European air quality by Heal et al. (2013) and Colette et al. (2011), respectively.





Another example is Chemel et al. (2014) which nests the WRF-CMAQ (Weather Research and
Forecasting - Community Multi-scale Air Quality) air quality model over the UK domain inside a
European regional model but takes initial and lateral boundary conditions (LBCs) for composition
and climate from two different global models. Some examples of future climate and air quality
simulations are those carried out by Trail et al. (2014), Meleux et al. (2007) and Langner et al.
(2012). Recognising the advantages of more closely-coupled meteorology and composition, online
models have increasingly been developed. Initially this was mainly in the context of global general
circulation models (GCMs) for climate modelling, where long time-scale simulations potentially
render even small feedback mechanisms between composition and meteorology important. Results
from some of these models have been used in the latest Intergovernmental Panel on Climate Change
(IPCC) Assessment reports (Boucher et al., 2013; Myhre et al., 2013; Lamarque et al., 2013). Online
regional chemistry models are a more recent development, with applications to air quality forecasting
(e.g. Savage et al., 2013; Baklanov et al., 2014) and impacts from a changing climate (e.g. Shalaby
et al., 2012; Colette et al., 2011; Forkel and Knoche, 2006). Hong et al. (2016), for example, nests
the online regional model WRF-CMAQ inside a different global model, CESM-NCSU (Community
Earth System Model - North Carolina State University). Single online chemistry models that can
be used at all scales, from global through regional and even to urban scale resolutions represent the
most advanced modelling configuration. The first model with this capability was GATOR-GCMM
(Gas, aerosol, transport, radiation, general circulation and mesoscale model, Jacobson, 2001) which
linked existing global and regional versions of the GATOR model such that the gas, aerosol and
radiative parts of the two scales were the same, although the meteorological and transport parts
differed. This capability has also since been implemented more recently in GU-WRF/Chem (Zhang
et al., 2012) which started from a mesoscale model (WRF/Chem) re-configured for the global scale.
These models are capable of running regional models nested within a consistent global chemistry
model.

In this paper we describe and evaluate a new modelling framework which uses a more consistent
set of models to go from the global scale down to the UK national scale. We employ the Met Of-
fice's Unified Model, MetUM (Brown et al., 2012), to downscale from a global composition-climate
model (GCCM) configuration to the UK national scale, via a regional composition-climate model
(RCCM) configuration. At each scale, model configurations of MetUM appropriate to the resolution
are employed, but the use of a single framework results in a higher degree of consistency across
the scales. The global climate model used is based on the Global Atmosphere 3.0 (GA3.0) config-
uration of HadGEM3 (Walters et al., 2011) and the RCCM is a limited area version, described by
Moufouma-Okia and Jones (2015). The inner nest is the regional air quality forecast model AQUM.
This operates at a resolution of 12 km and is used operationally to provide the UK national air qual-
ity forecast. The forecasts generated by AQUM are evaluated against hourly pollutant measurements
on a daily basis (Savage et al., 2013). Whilst we have sought to maximise consistency between the





models there do remain some differences and these are noted and described in subsequent sections. The purpose of the present paper is to describe the new modelling framework and to evaluate simulations of present day air quality by comparing against UK observations. The paper is structured as follows. Section 2 describes the modelling framework employed in this study. Section 3 describes

the experimental setup of the present-day simulations. Section 4 presents results on the performance of the nested configurations and a discussion with concluding remarks can be found in Section 5. This modelling framework has also been used to downscale global climate and air quality projections for the 2050s onto the UK national scale and is discussed in Folberth et al. (In prep.a).

## 2 Modelling System Description

In this section, we provide a brief overview of each of the scientific configurations of the MetUM employed in this study. We present a summary of the model dynamics, model physics, and details of the two-step, one-way nesting approach developed. A discussion of the chemistry and aerosol schemes is also included.

### 2.1 Global Composition-Climate Model (GCCM)

The GCCM is based on the Global Atmosphere 3.0/Global Land 3.0 (GA3.0/GL3.0) configuration of the Hadley Centre Global Environmental Model version 3 (HadGEM3, Walters et al., 2011), of the Met Office's Unified Model (MetUM, Brown et al., 2012). Soil-vegetation-atmosphere interactions are calculated using the Joint UK Land Environment Simulator (JULES, Best et al., 2011) and a full description of the GCCM can be found in Walters et al. (2011). The model has a horizontal

resolution of $1.875° \times 1.25°$, which translates to approximately $140 \times 140$ km in the mid-latitudes. The model has 63 levels in the vertical, spanning up to 41 km with the first 50 levels below 18 km. The model's dynamical time-step is 20 minutes.

The GA3.0 configuration of HadGEM3 (Walters et al., 2011) incorporates an interactive aerosol scheme, CLASSIC (Coupled Large-scale Aerosol Simulator for Studies in Climate, Jones et al.

(2001); Bellouin et al. (2011)). CLASSIC is a mass-based aerosol scheme in which all the aerosol components are treated as external mixtures. The scheme simulates ammonium sulphate, mineral dust, soot, fossil-fuel organic carbon (FFOC), biomass burning (BB) and ammonium nitrate in a prognostic (evolving) manner and biogenic secondary organic aerosols prescribed from a climatology. Sea salt is treated as a diagnosed quantity over sea points in the model; a limitation of this is

that it does not contribute to particulate matter predictions over land points. The aerosols can influence the atmospheric radiative and cloud properties through aerosol-radiation and aerosol-cloud interactions but for this study, these interactions have been switched off.

The (gaseous) chemistry in the GCCM is simulated by a tropospheric configuration of the United Kingdom Chemistry and Aerosol (UKCA) model (Morgenstern et al., 2009; O'Connor et al., 2014).



The large-scale transport of UKCA and CLASSIC tracers (and moisture variables) makes use of the
MetUM's dynamical core (Davies et al., 2005), with semi-Lagrangian advection and conservative
and monotone treatment of tracers (Priestley, 1993). Boundary layer mixing uses the scheme from
Lock et al. (2000) and includes an explicit entrainment parametrisation and non-local mixing in
unstable layers. Convective transport of tracers is based on the Gregory and Rowntree (1990) mass-
flux scheme, with more recent updates outlined in Martin et al. (2006). Physical removal of soluble
species is parametrised as a first-order loss process based on convective and stratiform precipitation
rates as described in O'Connor et al. (2014). Dry deposition is based on the resistance in-series
approach of Wesely (1989). The UKCA differential chemical equations are integrated in time using
an explicit iterative backward Euler approach (Hertel et al., 1993) with a chemical time-step of
5 minutes. Although UKCA has two options in relation to photolysis (O'Connor et al., 2014), the
photolysis reactions in this configuration are handled using offline rates, calculated in the Cambridge
2-D model (Law and Pyle, 1993) using the two-stream approach of Hough (1988). They are read in
by UKCA on the first time-step of the model integration and interpolated in time and space at each
model grid box. The impact of cloud cover, surface albedo and aerosols is included in the form of a
climatological cloud cover, prescribed albedo and aerosol loading, respectively. Note that although
the UKCA model has its own aerosol scheme (GLOMAP-mode, Mann et al., 2010), the CLASSIC
aerosol scheme has been used here, for consistency in the treatment of aerosols across the different
model configurations of the MetUM.

A detailed description of the UKCA tropospheric chemistry configuration can be found in O'Connor
et al. (2014). However, for this study, an extended tropospheric chemistry scheme, called UKCA-
ExtTC, which has been applied successfully in previous studies of tropospheric chemistry (e.g.,
Ashworth et al., 2012; Pacifico et al., 2015) has been employed. A separate, detailed description
of this extended version of UKCA is in preparation (Folberth et al., In prep.b). The UKCA-ExtTC
chemical mechanism has been designed to represent the key chemical species and reactions in the
troposphere in as much detail as is necessary to simulate atmospheric composition, air quality and
the interaction between atmospheric composition and climate while retaining the capability to con-
duct decade-long climate simulations. UKCA-ExtTC includes 89 chemical species, 63 of which are
transported as 'tracers'. For the remaining 26 species, transport is negligible in comparison to chem-
ical transformation during one model time-step and hence they are treated as 'steady-state' species.
UKCA-ExtTC uses the same backward Euler solver, chemical time-step (5 min), offline 2-D pho-
tolysis scheme and large-scale, convective transport and boundary layer treatment of tracers as the
scheme in O'Connor et al. (2014).

A two-way coupling between the ExtTC chemistry scheme and the CLASSIC aerosol scheme
is applied via the oxidant species (ozone($O_3$), the hydroxl (OH) and hydroperoxyl ($HO_2$) radicals,
hydrogen peroxide ($H_2O_2$) and nitric acid ($HNO_3$)) which drive the aqueous-phase oxidation of



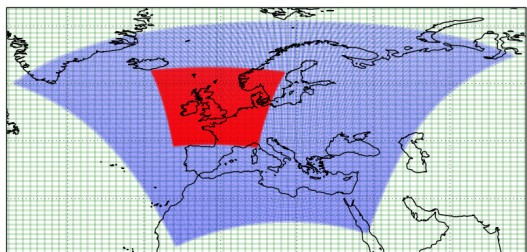

**Figure 1.** Nest of all three modelling domains. An extract of the GCCM is shown in green (the resolution of the model grid-boxes can be clearly seen). The RCCM domain is plotted in blue and AQUM in Red.

dimethyl sulphide (DMS) and sulphur dioxide ($SO_2$) to sulphate and ammonia ($NH_3$) to ammonium nitrate.

### 2.2 Regional Composition-Climate Model (RCCM)

The RCCM, referred to as the HadGEM3-A 'regional' (HadGEM3-RA) configuration, is described

in detail in Moufouma-Okia and Jones (2015), and is also based on the GA3.0/GL3.0 configuration of HadGEM3 (Walters et al., 2011). The RCCM has a horizontal resolution of $0.44° \times 0.44°$ (roughly $50 \times 50$ km) with a domain covering most of Europe and N. Africa (Figure 1) and the same 63 vertical levels as the GCCM. The RCCM closely follows the GCCM configuration (Section 2.1), with the same dynamical solver, radiation, precipitation and cloud (PC2) schemes. The same principal

components are included: the UKCA-ExtTC chemistry model, the CLASSIC aerosol model and the JULES land-surface model. The model dynamical time-step was reduced to 12 minutes (20 min in GCCM) to account for the increase in resolution and shorter turnaround of dynamical processes and interactions. The chemical time-step is 5 minutes. Boundary conditions, used to drive the RCCM from the GCCM, will be discussed in Section 3.

### 2.3 AQUM

The final, high resolution nest employed is the air quality forecast model AQUM (Air Quality in the Unified Model). AQUM, like both the GCCM and the RCCM, is also based on the MetUM. AQUM has a horizontal resolution of $0.11° \times 0.11°$ (approx $12 \times 12$ km) on a 'rotated pole' grid, covering the UK and nearby Western Europe (see Figure 1), with 38 vertical levels up to 39 km. The

LBCs, provided by the RCCM, are on 63 levels but interpolated onto the 38 levels of AQUM. The dynamical and chemistry time-steps are both 5 minutes.

The set-up of this model is described in detail in Savage et al. (2013) and uses the same parametrisation schemes as the Global and Regional CCMs described above, apart from large scale cloud, where AQUM uses the diagnostic cloud scheme as described by Smith (1990). As with the GCCM



and RCCM, AQUM uses the CLASSIC aerosol scheme (Jones et al. (2001); Bellouin et al. (2011)) and the UKCA model for its gas-phase chemistry. This helps to improve consistency between many aspects of the models. For example, large-scale and convective transport, boundary layer mixing, and wet and dry deposition are similar between all the nests. However a different chemistry mechanism, the Regional Air Quality (RAQ) scheme is used and the photolysis scheme also differs. Photolysis

rates in AQUM are calculated with the on-line photolysis scheme Fast-J (Wild et al., 2000; O'Connor et al., 2014), which is coupled to the modelled liquid water and ice content, and sulphate aerosols at every time step.

    The RAQ chemistry scheme pre-dates the ExtTC scheme and has been used in AQUM throughout its development and use as a forecast model. The experience developed with AQUM and the under-

standing of model performance established relies on the continuing use of this scheme and therefore we chose to retain this scheme for the final nest. The scheme has 40 transported species, 18 non-advected species, 116 gas-phase reactions and 23 photolysis reactions; 16 of the transported species are emitted: nitrogen oxide (NO), methane ($CH_4$), carbon monoxide (CO), formaldehyde (HCHO), ethane ($C_2H_6$), acetaldehyde ($CH_3CHO$), propane ($C_3H_8$), acetone ($CH_3COCH_3$), isoprene ($C_5H_8$),

methanol ($CH_3OH$), hydrogen ($H_2$), ethene ($C_2H_4$), propene ($C_3H_6$), butane ($C_4H_{10}$), toluene and o-xylene. As was the case in the GCCM and the RCCM, there is two-way coupling of oxidants between CLASSIC and the RAQ chemistry scheme. Further details of the RAQ scheme can be found in Savage et al. (2013).

### 3   Experimental Setup

In this section, a description of the experimental setup for modelling present-day air quality using the configurations of MetUM is provided, covering meteorological lower boundary conditions, emissions, upper boundary conditions, and lateral boundary conditions.

#### 3.1   Model Simulations and Model Calibration

    Both the GCCM and the RCCM were initialised using meteorological fields from a 20-year spin-up

of the standard HadGEM3 configuration. The model simulations for both these model configurations cover a total period of six years of which the first year is considered as spin-up and only the last five years are used in the analysis. The GCCM was used to produce the off-line lateral boundary conditions (LBCs) at six-hourly intervals to drive the RCCM, together with the emissions and upper and lower boundary conditions described below. LBCs include meteorological drivers (3D-winds, air

temperature, air density, Exner pressure, humidity and cloudiness), important chemical tracers from UKCA-ExtTC ($O_3$, NO, nitrogen dioxide ($NO_2$), $HNO_3$, dinitrogen pentoxide ($N_2O_5$), $H_2O_2$, $CH_4$, CO, HCHO, $C_2H_6$, $C_3H_8$, $CH_3COCH_3$, peroxy acetly nitrate (PAN)), gas-phase aerosol precursors ($SO_2$, DMS) and aerosols (dust, sulphate, nitrate, soot, FFOC and BB) from CLASSIC. In turn,





the RCCM produced meteorological and composition LBCs required to drive the national-scale air

quality model AQUM. Simulations with AQUM were initialised from the last month of the first year of the RCCM and were continued for five model years applying the LBCs supplied by the RCCM off-line at six-hourly intervals. The chemical and aerosol species provided in the LBCs are: Dust, $SO_2$, DMS, $SO_4$, Soot, OCFF, Nitrate, $O_3$, NO, $NO_2$, $N_2O_5$, $HONO_2$, $H_2O_2$, $CH_4$, CO, HCHO, $C_2H_6$, PAN and $C_3H_8$.

For lower boundary conditions the GCCM used monthly mean distributions of sea surface temperature (SST) and sea ice cover (SIC), derived for the present-day (1995-2005) from transient coupled atmosphere-ocean simulations (Jones et al., 2011) of the HadGEM2-ES model (Collins et al., 2011). The vegetation distribution for each of the simulations was prescribed using the simulated vegetation averaged for the same decade from this transient climate run, on which crop area, as given in

the 5th Coupled Model Intercomparison Project (CMIP5) land use maps (Hurtt et al., 2011; Riahi et al., 2007), was superimposed. The same present-day SST and SIC climatologies developed for the GCCM were downscaled to the RCCM and then AQUM domains using a simple linear regridding algorithm.

The GCCM was calibrated against $O_3$ measurements from the monitoring station located at Mace

Head Atmospheric Research Station in West Ireland at 53.3° North and 9.9° West. It is part of the Automatic Urban and Rural Monitoring Network (AURN) which is run by a number of institutions coordinated by Defra. The Mace Head monitoring station is representative of rural background conditions. Model output has been compared to the annual cycle of monthly mean $O_3$ which is based on a multi-year climatology of observed near-surface $O_3$ concentrations. The parameter $O_3$ surface dry

deposition was used to perform the calibration as the model shows very high sensitivity to this parameter. The model has been optimized to reproduce both the magnitude and seasonal cycle of $O_3$ at the Mace Head site in the model domain as closely as possible by varying the $O_3$ surface dry deposition flux within its uncertainties limits. An increase of the $O_3$ dry deposition by 20% yielded the best agreement, both with respect to $O_3$ monthly mean surface concentration and seasonal cycle, with

the observed climatology at the Mace Head station, which is representative of the $O_3$ background concentration in the lower troposphere.

As the RCCM uses the same code-base as the GCCM, this calibration is inherited by the former automatically. The model calibration has been applied to optimize consistency between the individual configurations in the global-to-national model nesting chain.

Due to the different chemistry scheme used in AQUM, the calibration used by the GCCM was not incorporated into AQUM as the RAQ scheme has been developed with performance over the UK as its main focus. This is unlike the GCCM where usually performance has to be taken into account over the entire globe which may lead to worse performance in some regions such as the UK.



### 3.2 Emissions

A consistent set of emissions has been used for all three model configurations through using the same source data, but then regridding to the required resolution for each model.

The emissions of reactive gases and aerosols from anthropogenic and biomass burning sources used in this study are based on the dataset used for Fifth Coupled Model Inter-comparison Project (CMIP5) simulations and described by Lamarque et al. (2010). The models are all driven by decadal

mean present-day emissions from CMIP5, representative of the decade centred on 2000. An example of the emissions for the different domains is given for NO in Fig. 2, while a full set of emission totals can be seen in Tables A1, A2 and A3.

UKCA-ExtTC takes into account emissions for 17 of its chemical species: nitrogen oxides ($NO_x$ =NO + $NO_2$), carbon monoxide (CO), hydrogen ($H_2$), methanol, formaldehyde, acetaldehyde and

higher aldehydes, acetone ($CH_3COCH_3$), methyl ethyl ketone, ethane ($C_2H_6$), propane ($C_3H_8$), butanes and higher alkanes, ethene, propene and higher alkenes, isoprene, (mono)terpenes and aromatic species. Of these butanes and higher alkanes, propene and higher alkenes, terpenes and aromatics are treated as lumped species. Surface emissions are prescribed in most cases. The only exception is the emission of biogenic volatile organic compounds (BVOCs) which are calculated interactively in

JULES using the iBVOC emission model (Pacifico et al., 2011). The emission of biogenic terpenes, methanol and acetone follows the model described in Guenther et al. (1995). As summarised in Table A2, global annual total emissions of biogenic isoprene and monoterpenes interactively computed with the iBVOC model of, for instance, 480 Tg(C) $yr^{-1}$ and of 95 Tg(C) $yr^{-1}$ are in reasonably good agreement with most other state-of-science interactive biogenic VOC emission models (e.g.,

Lathière et al., 2005; Guenther et al., 2006; Arneth et al., 2007; Müller et al., 2008; Messina et al., 2016) and global bVOC emission inventories (e.g., Arneth et al., 2008; Sindelarova et al., 2014). A detailed evaluation of the model performance is presented in Pacifico et al. (2011)

Emissions of $NO_x$ from lightning is taken into account in UKCA. Lightning $NO_x$ emissions are calculated interactively at every time step, based on the distribution and frequency of lightning

flashes following Price and Rind (1992, 1993, 1994). In this parametrisation the lightning flash frequency is proportional to the height of the convective cloud top in all the models. For cloud-to-ground (CG) flashes lightning $NO_x$ emissions are added below 500 hPa, distributed from the surface to the 500 hPa level, while $NO_x$ emissions resulting from intra-cloud (IC) flashes are distributed from the 500 hPa level up to the convective cloud top. The emission magnitude is related to the discharge

energy where CG flashes are 10 times more energetic than IC flashes (Price et al., 1997). The scheme implemented in the GCCM produces a total global emission source of around 7 Tg(N) $yr^{-1}$ which is in good agreement with the literature (c.f., e.g., Schumann and Huntrieser, 2007).

Soil-biogenic $NO_x$ emissions are taken from the monthly mean distributions from the Global Emissions Inventory Activity (http://www.geiacenter.org/inventories/present.html), which are based



**Table 1.** VOC split to convert total emitted VOCs from ExtTC to RAQ emitted VOCs. These factors sum to 1.0.

| Species | Conversion factor |
| --- | --- |
| HCHO | 0.055 |
| $C_2H_6$ | 0.156 |
| $CH_3CHO$ | 0.015 |
| $C_3H_8$ | 0.110 |
| $CH_3COCH_3$ | 0.078 |
| $CH_3OH$ | 0.116 |
| $C_2H_4$ | 0.079 |
| $C_3H_6$ | 0.034 |
| $C_4H_{10}$ | 0.238 |
| toluene | 0.095 |
| o-xylene | 0.024 |

on the global empirical model of soil-biogenic $NO_x$ emissions of Yienger and Levy II (1995) giving a global annual total of 5.6 Tg(N) $yr^{-1}$.

For $CH_4$, the UKCA model can be run by prescribing surface emissions or prescribing either a constant or time-varying global mean surface concentration. For the simulations being evaluated here, a time-invariant $CH_4$ concentration of 1760 ppbv was prescribed at the surface.

The sea salt and mineral dust emissions are computed interactively at each model time step based on instantaneous near-surface wind speeds (Jones et al., 2001; Woodward, 2001). Similarly the ocean DMS emissions are computed based on wind-speed, temperature and climatological ocean DMS concentrations from Kettle et al. (1999), using the sea-air exchange flux scheme from Wanninkhof (1992).

Emissions for AQUM are derived by re-gridding emissions from the regional model to the required $0.11°$ resolution. The ExtTC and RAQ chemistry schemes emit different anthropogenic VOC species, consequently some conversion is required. Our approach is to sum the anthropogenic VOC emission from ExtTC and apportion this total according to the values given in Table 1. These values were derived using the tabulated VOC emission fraction data over the UK for 2006 given by Dore

et al. (2008). For biogenic isoprene emissions, AQUM uses an off-line, monthly varying climatology which was derived from the on-line isoprene emission fluxes generated by the RCCM. A diurnal cycle is applied to account for daylight hours.

### 3.3 AQUM with higher resolution emissions

Following an initial evaluation of results, an additional model run was also carried out using AQUM.

This run was identical to the main AQUM run (using the same RCCM LBCs), with the exception of the anthropogenic emissions used. A new set of the latter were produced based on the higher



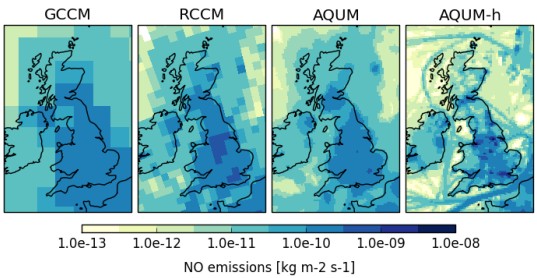

**Figure 2.** NO emissions for all models: GCCM (left), RCCM, AQUM and higher-resolution emissions run (AQUM-h) (right).

resolution datasets which AQUM uses for its operational air quality forecast; these are described further in Savage et al. (2013). Figure 2 shows the impact of these emissions for NO. The highest resolution input data to these emissions are at 1 km over the UK, although regridded to the 12 km

resolution required by AQUM. These are based on 2006 emissions, but the total emission has been rescaled to match the year-2000 decadal mean areal totals given by Lamarque et al. (2010) (as described in Section 3.2). For the remainder of the paper, this additional run will be referred to as AQUM-h.

### 3.4   Upper Boundary Conditions

While the chemistry is calculated interactively up to the model top in each configuration, upper boundary conditions are applied at the top of each model domain to account for missing stratospheric processes such as those related to $CH_4$ oxidation and bromine and chlorine chemistry. These boundary conditions are described in detail in O'Connor et al. (2014) and are only briefly discussed here. For $O_3$, the field used in the radiation scheme by MetUM in the absence of interactive chemistry

is used to overwrite the modelled $O_3$ field in all model levels that are 3–4 km above the diagnosed tropopause (Hoerling et al., 1993). For stratospheric odd nitrogen species ($NO_y$), a fixed $O_3$ to $HNO_3$ ratio of $\frac{1.0}{1000.0}$ $\frac{kg(N)}{kg(O_3)}$ from Murphy and Fahey (1994) is applied to $HNO_3$ in the same vertical domain. Finally, for $CH_4$, an additional removal term is applied in the three uppermost levels of the model. This $CH_4$ loss term was calculated in O'Connor et al. (2014) to be $50\pm10$ $TgCH_4yr^{-1}$ in a

global configuration.

### 4   Results

Our aim is to evaluate the air pollutant concentrations output from the RCCM and AQUM simulations using different datasets representative of the true air quality in the UK. In this way, we also aim to assess the potential for improving modelled air pollutant concentrations by increasing model





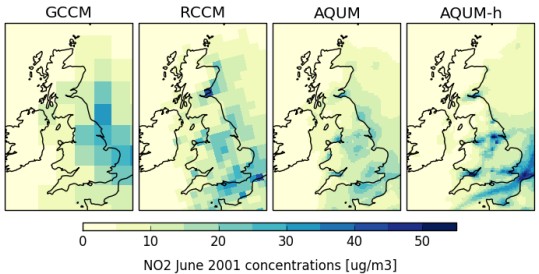

**Figure 3.** Monthly mean NO$_2$ concentrations over the UK for June for the four different model runs. From left to right: GCCM, RCCM, AQUM, AQUM-h.

spatial resolution. The datasets we use include (i) in situ observations of hourly air pollutant concentrations from the UK Automatic Urban and Rural Network (AURN) and (ii) annual mean surface pollutant concentrations produced by the Pollution Climate Mapping (PCM) model which also takes into account observations, described by Brookes et al. (2013). This model produces gridded fields at a spatial resolution of 1 km over the whole of the UK.

Another aspect of the analysis undertaken is to employ two different approaches to model assessment. The first uses standard verification metrics such as bias based on site-specific comparisons averaged over the five year modelled period. The second approach uses neighbourhood verification techniques which consider the area surrounding a particular point and thus allow for some mis-match in the spatial positioning of elevated pollutant values, thereby avoiding the well-known

'double penalty' problem (Mittermaier, 2014).

   We begin with a qualitative comparison of the GCCM against the two limited-area models in order to illustrate the need for improved resolution over that of the GCCM for air quality applications.

### 4.1 Comparison to GCCM

Figure 3 compares UK monthly mean NO$_2$ concentrations for June calculated from runs of the

GCCM, RCCM, AQUM and AQUM-h models. In the GCCM plot the resolution is wholly insufficient to realistically represent the elevated NO$_2$ levels around the UK urban centres (London, West Midlands, Greater Manchester, West Yorkshire, Edinburgh) and in the busiest shipping lanes and ports (English Channel, Bristol Channel, Southampton, Liverpool). The representation improves qualitatively as we move to the right in this plot. It can clearly be seen that higher resolution mod-

elling is essential for providing realistic pollutant representations at more localised spatial scales.

### 4.2 Comparison against in situ observations

In this section we compare results from the RCCM, AQUM and AQUM-h simulations with suitable averages derived from observations from the UK Automatic Urban and Rural Network (AURN,





**Table 2.** Statistics comparing modelled air pollutant concentrations to AURN observations, for the period 1st Jan 2001 - 31st Dec 2005.

|  |  | RCCM | AQUM | AQUM-h |
| --- | --- | --- | --- | --- |
| NO2 | Number of Sites | 65 | 65 | 65 |
|  | Bias ($\mu g m^{-3}$) | -4.76 | -5.47 | -0.80 |
|  | % Observations > Threshold (=65.0 $\mu g m^{-3}$) | 6.21 | 6.21 | 6.21 |
|  | % Model > Threshold (=65.0 $\mu g m^{-3}$) | 1.86 | 2.07 | 5.64 |
| $O_3$ | Number of Sites | 65 | 65 | 65 |
|  | Bias ($\mu g m^{-3}$) | 6.23 | 13.94 | 9.96 |
|  | % Observations > Threshold (=100.0 $\mu g m^{-3}$) | 2.39 | 2.39 | 2.39 |
|  | % Model > Threshold (=100.0 $\mu g m^{-3}$) | 3.18 | 8.54 | 7.07 |
| PM10 | Number of Sites | 40 | 40 | 40 |
|  | Bias ($\mu g m^{-3}$) | -12.45 | -13.32 | -14.41 |
|  | % Observations > Threshold (=50.0 $\mu g m^{-3}$) | 4.18 | 4.18 | 4.18 |
|  | % Model > Threshold (=50.0 $\mu g m^{-3}$) | 0.99 | 0.87 | 0.85 |
| PM2.5 | Number of Sites | 2 | 2 | 2 |
|  | Bias ($\mu g m^{-3}$) | 0.33 | -0.75 | -2.46 |
|  | % Observations > Threshold (=35.0 $\mu g m^{-3}$) | 1.08 | 1.08 | 1.08 |
|  | % Model > Threshold (=35.0 $\mu g m^{-3}$) | 3.93 | 3.11 | 2.40 |
| $SO_2$ | Number of Sites | 49 | 49 | 49 |
|  | Bias ($\mu g m^{-3}$) | 2.61 | 1.44 | 1.59 |
|  | % Observations > Threshold (=25.0 $\mu g m^{-3}$) | 2.89 | 2.89 | 2.89 |
|  | % Model > Threshold (=25.0 $\mu g m^{-3}$) | 3.98 | 3.71 | 5.31 |

https://uk-air.defra.gov.uk/networks/network-info?view=aurn) for 2001-2005. From this network we

only consider 'background' sites which include the site classifications of remote, rural, suburban and urban background. We are therefore excluding sites which we expect to be un-representative of a large area, such as roadside or industrial sites. As the models are driven by climatological meteorology, we do not expect the model results to match the hourly AURN observations, hence we compare values averaged over the five year period with corresponding averages derived from the

hourly observations.

### 4.2.1 NO$_2$

Figure 4(a) shows a frequency distribution of hourly observed concentrations of NO$_2$ with corresponding frequency distributions for modelled concentrations from the RCCM, AQUM, and AQUM-h configurations. It is clear that the AQUM-h model distribution more closely matches the observed

distribution than the other model configurations, illustrating the importance of increased spatial resolution and emissions for this pollutant. Corresponding statistical measures of model skill are given





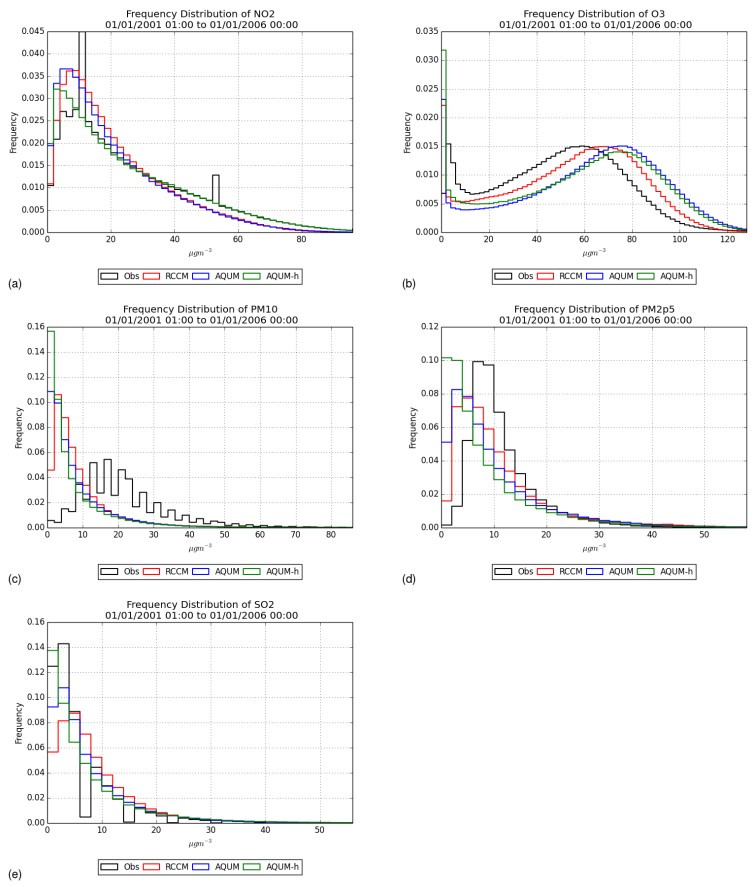

**Figure 4.** Frequency distribution of main pollutants: (a) $NO_2$, (b) $O_3$, (c) PM10 (d) PM2.5 and (e) $SO_2$. Observations are shown in black, RCCM in red, AQUM in blue and AQUM-h in green.

in Table 2. The bias in RCCM and AQUM against AURN observations is -4.76 and -5.47 $\mu g m^{-3}$, respectively, but is reduced to -0.80 $\mu g m^{-3}$ in AQUM-h. In Table 2 a comparison of the percentage of observations/model values greater than the 65.0 $\mu g m^{-3}$ threshold is also included; it illustrates

that AQUM-h simulates observed frequencies of higher $NO_2$ concentrations well, making it better suited to calculate health burdens due to elevated levels of $NO_2$ (e.g. Pannullo et al., 2017). However shown in Fig. 5(a) is a comparison of the seasonal cycle of observed and modelled $NO_2$ concentrations, averaged over all AURN sies considered. This shows that none of the models are able to fully capture the seasonal cycle of $NO_2$, with wintertime modelled concentrations biased low, while the

RCCM and AQUM straddle the observed concentrations during summer. This is likely to be due to





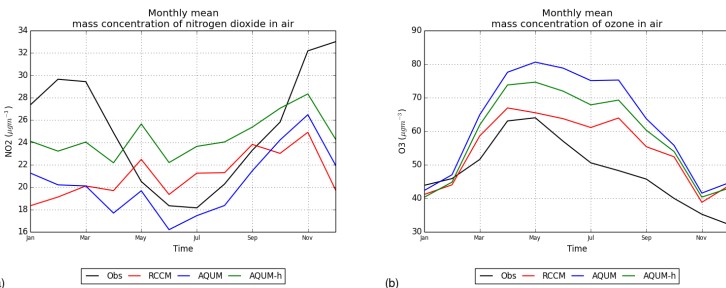

**Figure 5.** Monthly mean concentrations of (a) $NO_2$ and (b) $O_3$. Observations are shown in black, RCCM in red, AQUM in blue and AQUM-h in green.

the poor representation of the monthly variation of emissions over the UK in the global model which is then inherited by the higher resolution models.

### 4.2.2 $O_3$

Relevant statistics are given in Table 2, while a frequency distribution plot, showing the distribution
of hourly $O_3$ concentrations over the entire period for models and observations, is shown in Fig. 4(b) and the seasonal cycle is given in Fig. 5(b). The latter plot illustrates that the pattern of the seasonal cycle of $O_3$ is captured reasonably well however the modelled spring/summer maximum persists too long and does not replicate the gradual decline in monthly mean concentrations as indicated by observations. This has implications for the use of modelled $O_3$ to quantify health impacts
from long-term exposure to $O_3$ during warmer months, as indicated by studies in North America (WHO, 2013a; COMEAP, 2015). In the frequency distribution plots in Fig. 4(b), it can be seen that all models are able to reproduce the shape of the observed distribution quite well but differ in their most frequent concentration, corresponding to different model biases. The RCCM exhibits the smallest bias against observations of +6.23 $\mu g m^{-3}$ and AQUM the greatest at +9.96 $\mu g m^{-3}$(see
Table 2). However the RCCM used an off-line photolysis scheme (O'Connor et al., 2014) whilst both configurations of AQUM used the interactive Fast-J scheme (Wild et al., 2000). Given the different photolysis schemes used, a sensitivity experiment for a single month of July was carried out, in which AQUM-h was re-run with off-line photolysis. The $O_3$ bias for this month is 7.33 $\mu g m^{-3}$ for the RCCM, 22.48 $\mu g m^{-3}$ for AQUM and 13.95 $\mu g m^{-3}$ for AQUM-h. Running AQUM-h with
the off-line scheme brings the bias down to 6.99 $\mu g m^{-3}$ which is marginally better than the RCCM. The sensitivity of surface $O_3$ to the choice of photolysis scheme found here, however, differs from two previous studies (O'Connor et al., 2014; Telford et al., 2013). Both of these studies found that $O_3$ decreased in the northern hemisphere by less than 5% when switching from off-line to on-line photolysis and indeed, the changes in the tropospheric $O_3$ budget were consistent between the two
studies. In addition, O'Connor et al. (2014) found no significant change in modelled $O_3$ evident at



NH mid-latitude sites (e.g. Mace Head). However both O'Connor et al. (2014) and Telford et al. (2013) were global studies rather than the regional scale considered here. Another conflicting factor is the calibration which has been applied to the RCCM for the $O_3$ dry deposition which would have an impact on the $O_3$ concentrations, although this would have impacted AQUM through the LBCs.

This calibration was not included in the papers described above which may help to explain the conflicting results. Consequently, these factors make it difficult to isolate and quantify the impact of the higher resolution third nest on model performance.

### 4.2.3 PM10

Relevant statistics are given in Table 2, while a frequency distribution plot, showing the distribution

of hourly PM10 values over the entire period for models and observations, is shown in Fig. 4(c).

For PM10, none of the models are able to reproduce the shape of the observed distribution and there is a significant negative bias across all the model configurations (between -12.45 and -14.41 $\mu g m^{-3}$). The lack of sea salt in modelled values over land points plays a significant role in this under-prediction. However poor modelling performance for PM10 is a common feature of many

global composition and regional air quality models (e.g. Colette et al., 2011; Im et al., 2015) and is often attributed to the unreliability of emissions of coarse component aerosol. This could potentially affect the quantification of health effects due to short-term and long-term exposure of PM10, as documented by WHO (2013a).

### 4.2.4 PM2.5

Relevant statistics are given in Table 2, while a frequency distribution plot, showing the distribution of hourly PM2.5 values over the entire period for models and observations, is shown in Fig. 4(d).

For the finer PM2.5 component of aerosol, the models perform significantly better in capturing the shape of the observed distribution than for PM10; there is a small positive bias for PM2.5 in the RCCM (+0.33 $\mu g m^{-3}$), whereas AQUM becomes slightly negative (-0.75 $\mu g m^{-3}$) and AQUM-h

more negative still (-2.46 $\mu g m^{-3}$).

However the observed frequency distribution is only based on 2 background observational sites available for PM2.5 in the UK for the 2001-2005 time period. The introduction of PM2.5 monitoring stations in the UK increased significantly from 2009 and we explored the possibility of using observations from 2011-2015 to generate a proxy for the 2001-2005 frequency distribution. However

we found that the PM10 distribution changed significantly over the 10 years and concluded that it was not valid to use the more recent PM2.5 observations in place of 2001-2005 observations. Consequently, due to the paucity of PM2.5 observations for the 2001-2005 time period against which to compare, for the remainder of this paper, we shall no longer consider PM2.5 results.

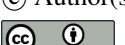



### 4.2.5 SO$_2$

Relevant statistics are given in Table 2, while a frequency distribution plot, showing the distribution of hourly SO$_2$ values over the entire period for models and observations, is shown in Fig. 4(e).

For SO$_2$, the model configurations exhibit similar distributions to the observed distribution, with generally positive biases of between +1.44 and +2.61 $\mu g m^{-3}$.

### 4.3 Comparison against PCM

In order to assess the variation in the quality of modelled air pollutant concentrations between the different model configurations, it is necessary to consider full spatial fields rather than the site comparison afforded by in situ observations described in the preceding section. Therefore, it is essential to compare the models against a realistic spatial field and for this purpose, we use fields derived from the Pollution Climate Mapping (PCM) model, as described in Brookes et al.

(2013). This sophisticated model combines information from a variety of sources, including emission inventories and observations datasets, to produce estimated annual mean surface pollutant concentrations on a 1x1 km grid over the entire UK for NO$_2$, SO$_2$, PM10 and PM2.5. The data are freely available at https://uk-air.defra.gov.uk/data/pcm-data. These results are widely used in the UK to provide the background pollutant concentrations for local air quality modelling stud-

ies and new site impact assessment studies. O$_3$ is also modelled by PCM but the output available is the number of days exceeding 120 $\mu g m^{-3}$, (as required by the European Union ambient air quality directives (http://eur-lex.europa.eu/LexUriServ/LexUriServ.do?uri=OJ:L:2008:152: 0001:0044:EN:PDF)) rather than pollutant concentrations and so cannot be used in our analysis. In view of the lack of AURN PM2.5 observations (also used in deriving the PCM maps) during the

period 2001-2005 (as described in Section 4.2.4) we have not considered PM2.5 in the following analysis.

PCM data for NO$_2$ and PM10 are available for 2001-2005, while SO$_2$ data are only available from 2002 onwards. A comparison (not shown) of the PCM against the in situ AURN observations as done for the models in Section 4.2 proved the PCM verifies better than any of the other models.

PCM Data from the available years were processed to produce five-year means (four-years for SO$_2$) for comparison with the similarly averaged model fields.

Comparisons between MetUM modelled annual mean concentrations and PCM annual mean concentrations are shown for NO$_2$, SO$_2$ and PM10 in Fig. 6. In these plots nearest neighbour regridding is used to interpolate the model fields and the PCM fields onto the 12 km AQUM grid. Spatial corre-

lations have been calculated between the regridded model and PCM fields (only at valid PCM data points, i.e UK land points) and are shown at the top of each figure.

For the primary pollutants of NO$_2$ (Fig. 6(a)) and SO$_2$ (Fig. 6(b)), there is an improvement in correlation with the PCM as we move from the RCCM to AQUM and finally AQUM-h: for NO$_2$ the





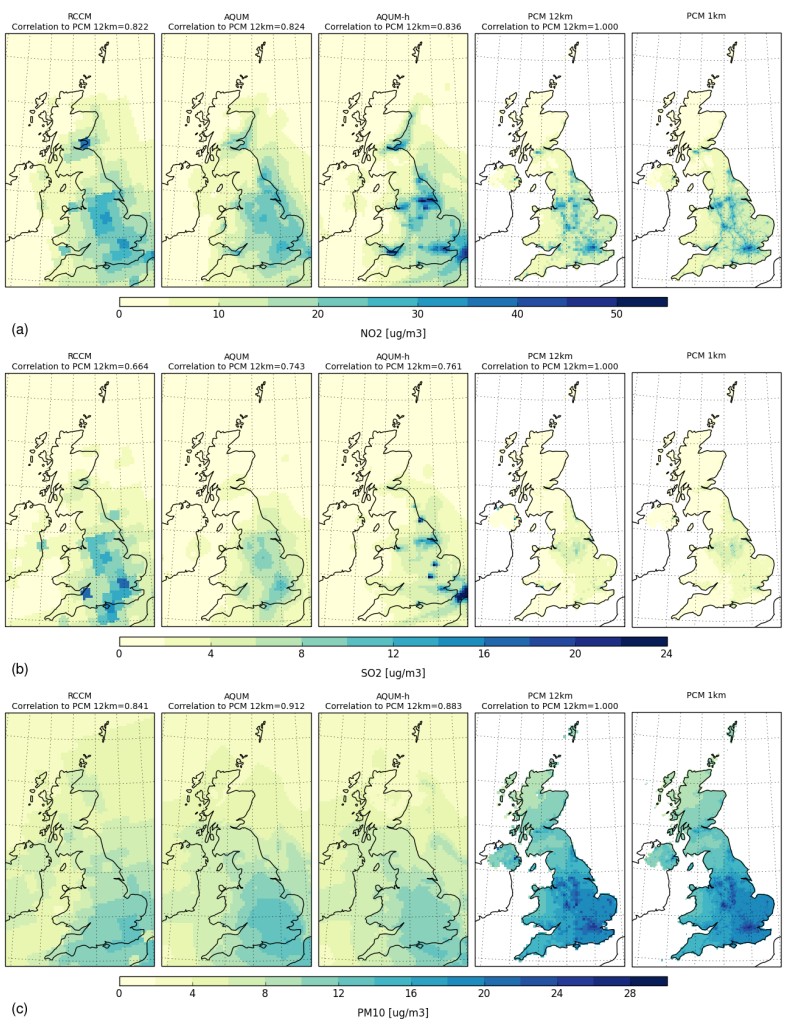

**Figure 6.** Model and PCM meaned fields for different pollutants, regridded onto 12 km AQUM grid. From left to right the models are RCCM, AQUM, AQUM-h, 12 km version of the PCM and finally the 1 km PCM for comparison. Plots also show the correlation between the fields and the 12 km version of the PCM. Pollutants shown are (a) $NO_2$ (top row), (b) $SO_2$ (middle) and (c) PM10 (bottom).

correlations are 0.822, 0.824 and 0.836, respectively, while for $SO_2$ the correlations are 0.664, 0.743
and 0.761, respectively. For $SO_2$, the introduction or removal of strong point sources can influence
the comparison via a calculated spatial correlation. This is apparent in the AQUM-h plots in Fig.
6(b), where two new strong point sources in south-eastern England are present in the 2006 data used
to generate the AQUM-h emissions. These modest increases of correlation with PCM (as our proxy





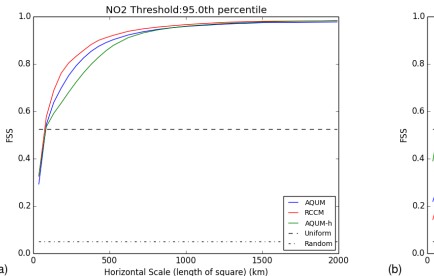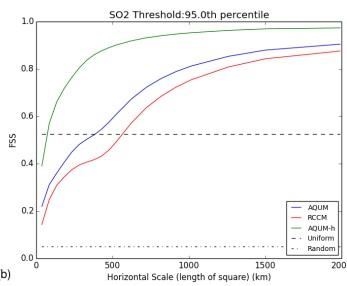

**Figure 7.** Fractional Skill Score for 95th Percentile for (a) NO$_2$ and (b) SO$_2$. The RCCM is shown in red, AQUM in blue and AQUM-h in green. The 'Random' (dot-dashed) line represents the FSS for a random forecast with the same fraction of points over the domain exceeding the percentile threshold as the truth field. The 'Uniform' (dashed) line represents a forecast with the same fraction of points above the percentile threshold in the neighbourhood surrounding each grid point as the truth field for every grid point. Above this line the forecast is considered skilful.

for 'truth') as model resolution increases, illustrate the benefits of increased resolution modelling,
both with respect to the model grid and the underlying emissions data, in better capturing the strongly inhomogeneous spatial distribution of these pollutants.

For PM10 however (Fig. 6(c)), this improvement in correlation with higher resolution is not as clear. The correlation values with the PCM are 0.841 for the RCCM, 0.912 for AQUM and 0.883 for AQUM-h. PM10 has a large secondary contribution which contributes a relatively smoothly varying
background to the PCM maps in Fig. 6(c). This is likely to be the reason for the lack of a clear improvement in PM10 modelling with the high resolution AQUM-h model.

Beyond the figures shown above, we also investigated the correlation scores by just considering data above fixed threshold concentration values (plots not shown). However these results were very variable, depending on the threshold values considered, partly due to the biases (as given in Section
495  4.2).

### 4.4 Analyses based on neighbourhood comparisons: the Fractional Skill Score

In evaluating a comparison of modelled air pollutant concentrations against some gridded representation of true concentrations (such as the PCM fields described above), small offsets in the spatial location of elevated values can give an exaggerated contribution to simple metrics such as bias and
root mean square error evaluated at each grid point. This is commonly referred to as the 'double penalty' problem. The resulting analysis may then give a misleading indication of the comparison between the two fields. So-called 'neighbourhood' verification techniques (Ebert, 2008; Mittermaier, 2014) have been developed to avoid these problems. Here, we consider the use of the Fractional Skill Score (FSS) (explained in detail in Roberts and Lean (2008)) to analyse the variation in model skill





in representing spatial patterns. This statistic has mainly been employed in evaluating the improve-
ments offered by high resolution precipitation forecasts, where a 'double penalty' problem occurs if
rain is forecast in a neighbouring grid box to where it was actually observed (hence an incorrect fore-
cast in both grid boxes). A lower resolution forecast might place the forecast and observed shower in
the same grid box, resulting in an apparently improved forecast. Similar issues are found in pollution

modelling due to the high degree of inhomogeneity of air pollutant concentrations and evaluation of
the FSS may offer improved comparisons.

The FSS is calculated by computing, for each grid box, the fraction of neighbouring grid boxes
which exceed a given threshold value (or percentile). This is done both for the gridded model fields
that are to be evaluated and a gridded benchmark field representative of the 'truth', which in this case

is the PCM fields, as described in Section 4.3. This can be repeated for varying neighbourhood sizes.
As the size of the neighbourhood increases, the fractional skill score should increase towards unity.
A forecast may be considered 'skilful' at the grid-scale where the model has the correct fraction of
points above the percentile threshold in the neighbourhood surrounding each grid point as the truth
field for every grid point

We have calculated the FSS using output from the 3 model configurations (RCCM, AQUM, and
AQUM-h) and compared to the PCM for various threshold values, based on both fixed thresholds
and percentile values. An example set of results is shown in Fig. 7. In these plots, the variation of
FSS against spatial scale is shown for the RCCM, AQUM and AQUM-h, using a 95th percentile
threshold. For $NO_2$, there is little difference between the three model configurations and the same

is found for PM10 (not shown). Calculations using other fixed thresholds and different percentile
thresholds also show little difference. However, for $SO_2$, AQUM-h shows the best performance,
crossing the threshold value of 0.5 at the shortest spatial scale and reflects the strong point sources of
$SO_2$ in contrast to $NO_2$ emissions. The use of neighbourhood verification techniques to compare our
different nests has therefore not offered any obvious increased insight into the differences between

the models and the consequent impacts on improved predictions across the spatial scales. This may
be an indication that the resolution differences between the models may not be the key factor in
determining performance, particularly for $NO_2$ and PM10.

## 5   Summary and Conclusions

This study describes the initial development of a more consistent framework for dynamic downscal-

ing of climate and air quality from a global composition-climate model to the national scale, via a
regional composition-climate model and thence to a higher resolution regional air quality forecast
model. In this attempt, some of the difficulties in presenting a clear-cut, quantitative demonstration
of the value of higher resolution modelling have been made apparent. All three models use a single
modelling framework - the MetUM - but some differences between the models do remain. The most

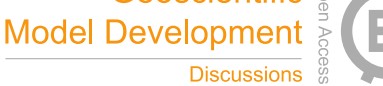



notable of these are the different chemistry mechanisms, photolysis schemes and the calibration factor that have been used in the GCCM and RCCM compared to AQUM. AQUM has been developed with forecasting air quality over the UK as its primary aim, and performance has been optimised for predicting in situ UK observations on an hourly timescale with a focus on high impact, more extreme events. By contrast, the GCCM and RCCM have been developed to predict global and regional climatologies, giving a faithful representation of seasonal and annual means across the entire globe. These differences have resulted in some of the inconsistencies highlighted in this paper. This has led to a challenge in determining the benefits of a three-level nest for downscaling to the regional scale but has highlighted important areas for consideration in future work.

The comparison of modelled air pollutant concentrations against in situ UK observations was conducted initially by a traditional site-specific analysis, with standard metrics such as bias. In addition, the impacts of model resolution on pollutant spatial patterns were assessed via comparison to the gridded PCM annual average pollution maps. In order to guard against the susceptibility of the traditional verification methods to the double penalty problem, an analysis was also carried out using a neighbourhood approach, utilising the Fractional Skill Score (FSS), although the results from this were generally inconclusive.

For $NO_2$, significantly improved modelled concentrations can be quantitatively demonstrated for the higher resolution models, using higher resolution emissions (biases of -4.76, -5.47 and -0.80 $\mu g m^{-3}$ for RCCM, AQUM and AQUM-h respectively. This is readily understood, given the dependence of surface concentrations of this primary pollutant on local emissions. For another primary pollutant, $SO_2$, a modest benefit of high resolution modelling is demonstrated by the small increase in spatial correlation of AQUM-h with the PCM climatology maps (correlations compared to the PCM of 0.664, 0.743 and 0.761 for RCCM, AQUM and AQUM-h). However the benefit is less pronounced for $SO_2$ than for $NO_2$ for two reasons: (i) in the UK, $SO_2$ levels have fallen dramatically over the last 25 years and ambient concentrations are now generally the result of relatively low magnitude traffic emissions and much stronger emissions from a small number of industrial point sources. This results in an annually averaged mean concentration map over the UK which shows relatively little spatial structure, but with a small number of locations having much higher concentrations due to strong local emission sources (see the PCM 1 km plot in Fig. 6(b)). This low level background with little overall spatial structure limits the quantitative increases in spatial correlation with the PCM climatologies. The other reason is related to the introduction and removal of strong point emissions sources affecting the comparison, as noted in section 4.3.

Conclusions regarding the benefits of high resolution modelling for PM2.5 have been hampered in the present study due to the lack of observations over the study period. This pollutant consists of both primary and secondary contributions and one might expect improvements in the modelling of the primary component by higher resolution modelling. However, the magnitude of the improvement will depend on the relative sizes of primary and secondary components and it may well be that the

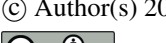



contribution of the large secondary component masks any improvement in the representation of the primary component. For PM10, model performance remains poor regardless of model resolution, with all three regional models (RCCM, AQUM and AQUM-h) failing to capture the observed frequency distribution having negative biases in the range -14.41 to -12.45 $\mu gm^{-3}$. The lack of the sea salt contribution to modelled PM10 estimates is a significant limitation; other important factors include the poor representation of other coarse component emissions and poor modelling of the growth of aerosols to sizes in the coarse range.

For $O_3$, all regional models were able to reproduce the shape of the observation distribution well, but the offset of the modelled from the observed central location varied. Tests showed that the differences are likely to be largely due to differences in the photolysis schemes employed. However, given the modest benefits of higher resolution modelling found for the other secondary pollutants it seems unlikely that high resolution modelling with AQUM would offer significantly improved performance for $O_3$ predictions beyond those demonstrated by RCCM.

The model simulations described in this paper have been evaluated in their air quality performance under present day climate. However, the same techniques can be applied for projecting future climate and air quality from the global scale to the UK national scale (Folberth et al., In prep.a). The ability to model air quality parameters at the regional scale will be particularly important for health impact modelling where high spatial resolution is important to allow the concentration variations to be matched to population locations. Indeed the techniques in this paper have already been applied to 2050s climate and air quality in Pannullo et al. (2017) for assessing potential changes in UK hospital admissions.

## 6 Code availability

Due to intellectual property right restrictions, we cannot provide either the source code or documentation papers for The Met Office's Unified Model, MetUM. The MetUM is available for use under licence. A number of research organisations and national meteorological services use the MetUM in collaboration with the Met Office to undertake basic atmospheric process research, produce forecasts, develop the MetUM code and build and evaluate Earth system models. For further information on how to apply for a licence see http://www.metoffice.gov.uk/research/modelling-systems/unified-model. JULES is available under licence free of charge. For further information on how to gain permission to use JULES for research purposes see https://jules.jchmr.org/software-and-documentation.

## Appendix A

Given in Tables A1,A2 and A3 are summaries of emission totals.



**Table A1.** Summary of the annual total emissions of trace gases used in the GCCM, RCCM and AQUM models.

| Species | GCCM | RCCM | AQUM |
|---|---|---|---|
| **$NO_x$ as Tg(N) yr$^{-1}$** | **49.4** | **8.1** | **2.3** |
| anthropogenic | 26.5 | | |
| forest/grassland fires | 4.3 | | |
| shipping | 5.5 | | |
| soil | 5.6 | | |
| lightning | 7.5 | | |
| **CO as Tg(CO) yr$^{-1}$** | **1112.8** | **85.2** | **20.2** |
| anthropogenic | 607.5 | | |
| forest/grassland fires | 459.1 | | |
| shipping | 1.2 | | |
| oceanic | 45.0 | | |
| **$CH_4$ as ppbv[a]** | **1760** | **1760** | **1760** |
| **$H_2$ as Tg(H$_2$) yr$^{-1}$** | **28.9** | **0.6** | **0.06** |
| forest/grassland fires | 28.9 | | |

[a] $CH_4$ surface concentration of 1760 ppbv is prescribed at the lower-most
model level;

*Acknowledgements.* The development of HadGEM3, UKCA, and the work of MD, GF, and FMO'C were sup-
ported by the Joint UK BEIS/Defra Met Office Hadley Centre Climate Programme (GA01101). LSN, MD, GF,
RNM and PA also acknowledge the Engineering and Physical Sciences Research Council (EPSRC) for addi-
tional funding through the UK Engineering and Physical Sciences research council grant number EP/J017485/1:
"A rigorous statistical framework for estimating the long-term health effects of air pollution". In addition, this
work and its contributors (GF and FMO'C) were partly supported by the UK-China Research & Innovation
Partnership Fund through the Met Office Climate Science for Service Partnership (CSSP) China as part of the
Newton Fund.

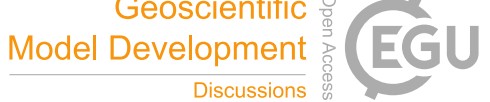



**Table A2.** Summary of the annual total emissions of volatile organic compounds used in the GCCM, RCCM and AQUM models.

| Species | GCCM | RCCM | AQUM | | GCCM | RCCM | AQUM |
|---|---|---|---|---|---|---|---|
| **$C_2H_6$ as Tg(C) yr$^{-1}$** | **5.4** | **0.3** | **0.4** | **$CH_3CHO$[b] as Tg(C) yr$^{-1}$** | **4.8** | **0.1** | **0.04** |
| anthropogenic | 2.6 | | | forest/grassland fires | 4.8 | | |
| forest/grassland fires | 2.6 | | | **$CH_3C(O)CH_3$** | | | |
| shipping | 0.2 | | | **as Tg(C) yr$^{-1}$** | **1.2** | **0.1** | **0.05** |
| **$C_3H_8$ as Tg(C) yr$^{-1}$** | **4.7** | **0.8** | **0.3** | anthropogenic | 0.2 | | |
| anthropogenic | 2.8 | | | forest/grassland fires | 1.0 | | |
| forest/grassland fires | 1.6 | | | **$CH_3C(O)CH_2CH_3$[c]** | | | |
| shipping | 0.3 | | | **as Tg(C) yr$^{-1}$** | **1.5** | **0.1** | **0.0** |
| **$C_4+$ alkanes as Tg(C) yr$^{-1}$** | **24.7** | **4.6** | **0.1** | anthropogenic | 0.2 | | |
| anthropogenic | 23.3 | | | forest/grassland fires | 1.3 | | |
| forest/grassland fires | 0.6 | | | **aromatics[d] as Tg(C) yr$^{-1}$** | **17.8** | **2.2** | **0.04** |
| shipping | 0.8 | | | anthropogenic | 13.8 | | |
| **$C_2H_4$ as Tg(C) yr$^{-1}$** | **16.5** | **1.1** | **0.2** | forest/grassland fires | 3.7 | | |
| anthropogenic | 9.4 | | | shipping | 0.3 | | |
| forest/grassland fires | 6.8 | | | **biogenicVOC** | | | |
| shipping | 0.3 | | | **as Tg(C) yr$^{-1}$** | **680** | | **0.2** |
| **$C_3+$ alkenes[a] as Tg(N) yr$^{-1}$** | **6.4** | **0.3** | **0.02** | as isoprene | 480 | | 0.2 |
| anthropogenic | 2.7 | | | as (mono-)terpenes | 95 | | 0 |
| forest/grassland fires | 3.4 | | | as methanol | 85 | | 0 |
| shipping | 0.3 | | | as acetone | 20 | | 0 |
| **HCHO as Tg(C) yr$^{-1}$** | **3.6** | **0.2** | **0.06** | | | | |
| anthropogenic | 1.3 | | | | | | |
| forest/grassland fires | 2.3 | | | | | | |

[a]includes $C_3$ plus higher alkenes and all volatile alkynes; [b]includes higher aldehydes; [c]includes methyl ethyl ketone (MEK) plus higher ketones; [d]includes benzene, toluene, and xylenes.





**Table A3.** Summary of the annual total emissions of aerosols used in the GCCM, RCCM and AQUM models.

| Species | GCCM | RCCM | AQUM |
|---|---|---|---|
| **black carbon (BC) as Tg(BC) yr$^{-1}$** | **5.8** | **0.9** | **0.2** |
| anthropogenic | 5.8 | | |
| shipping | 0.03 | | |
| **organic carbon (OC) as Tg(OC) yr$^{-1}$** | **13.5** | **1.9** | **0.2** |
| anthropogenic | 13.1 | | |
| shipping | 0.4 | | |
| **NH$_3$ as Tg(N) yr$^{-1}$** | | **7.1** | **1.7** |
| anthropogenic | 39.9 | | |
| forest/grassland fires | 3.1 | | |
| **SO$_2$ as Tg(SO$_2$) yr$^{-1}$** | **60.1** | **11.5** | **0.2** |
| anthropogenic | 49.1 | | |
| forest/grassland fires | 6.8 | | |
| shipping | 4.2 | | |





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
