# Peer review of "A description and evaluation of an air quality model nested within global and regional composition-climate models using MetUM"

_Geoscientific Model Development, 2017_

## Referee Comment (RC1) · Anonymous Referee #1 · 8 May 2017

The manuscript by Neal et al. presents the results of new simulations using an air quality model nested within a regional composition-climate model, which, in turn, is nested within a global model. The unique aspect of this work is the high level of consistency between the different models involved. Exceptions also exist, such as the different chemical mechanisms, photolysis treatments, and calibration factors used in the models. These inconsistencies, however, offer some opportunity for further insight. The model is described in detail, and the results for NO2, ozone, SO2, and PM are evaluated against different datasets. A neighbourhood approach is also used to assess the impact of the "double penalty problem" on the comparison.

The manuscript is clear, well written, and within the scope of the GMD journal. The

methodology used is sound, and the conclusions well discussed (although in some cases that I mention below, a bit more insight would be helpful). Therefore, I suggest its publication following the minor revisions and clarifications suggested below.

SPECIFIC COMMENTS:

Page 1, Line 2: I suggest appending "simultaneously" at the end of this sentence.

Page 1, Lines 14-15: "derived from a combination of modelling and observations" -> be more specific, does this refer to reanalysis data, and if so, which?

Page 1, Line 19: Please subscript "10".

Page 1, Line 21: "consistency between nested models is also important" -> This looks like a key conclusion, so I suggest being more explicit about what is meant. In fact, after reading the manuscript I am not sure how the results presented in it lead to this conclusion. Can the authors please explain or remove?

Page 2, Line 26: I suggest changing "constituents" to "pollutants", as otherwise it is a bit unconventional to mention ozone before $CO_2$ and methane as important for climate.

Page 2, Lines 44-45: Not clear what is meant – are they more inhomogeneous than, e.g. cloud distributions?

Page 2, Lines 46-47: Not sure why the second part of the sentence in the parenthesis is relevant here.

Page 5, Lines 140-143: Why has the 2D scheme been chosen instead of the more detailed approach? Is it due to the computational cost?

Page 5, Lines 145-148: Perhaps this sentence would seem better placed earlier on, when the use of CLASSIC is first mentioned. Again, are UKCA aerosols not chosen because of computational cost?

Page 5, Line 157: Is 89 a higher or a lower number compared to what is used in the

standard (non-ExtTC) UKCA?

Page 5, Line 164: Please add space after "ozone". Also, I suggest adding "simulated" before "oxidant species".

Fig. 1: I suggest either removing the more coarsely distanced lines from the map, to avoid confusion, or to add tick marks next to them. I also suggest that the resolution of the different domains is mentioned again in the caption.

Page 7, Line 214: Why were 20 years of spin-up needed?

Page 8, Line 251: I suggest adding "in the focus area" at the end of this sentence.

Page 8, Lines 255-258: Does this imply that the calibration described just above will not mean much for AQUM performance? That is a bit confusing.

Page 11, Line 321: The fact that the simulation is for average conditions around year 2000 has been elusive throughout the Experimental Setup section. I suggest clarifying this at the very beginning of the section.

Fig. 2: Why would some spatial structure existing in RCCM over central Britain disappear when moving to AQUM? Also, please place "-2" and "-1" in superscripts.

Page 13, Line 364: I am confused again – why 2001-2005 while earlier it was mentioned that the runs are designed for being representative of $\sim$2000?

Fig. 4: Please subscript "2", "3", "10" etc. in pollutant chemical formulas/abbreviations (also in other parts of the text), and change "2p5" to "2.5".

Page 14, Line 383: "sies" -> "sites".

Page 15, Line 386: Maybe I am wrong, but wouldn't the seasonality of emissions over the UK be similar across the different scales? Not sure why the global model in particular would have an issue with the seasonality of emissions.

Page 15, Lines 404-405: So, the photolysis scheme seems responsible. The discus-

sion below is useful, but maybe some further insight would be required here given this counter-intuitive behaviour, i.e. a more detailed model performing less well. At least some basic insight on whether key photolysis reactions for ozone (NO2, O3->O1D) become faster or slower?

Page 16, Lines 422-423: However, the more focused domain performs even worse, which should be mentioned.

Page 17, Lines 447-448: And AQUM performs somewhat better – worth mentioning. Also, the fact that SO2 performs ok is some (admittedly not so solid) indication that sulphate may not be the main contributor to PM biases? May be worth considering, in order to provide a bit more insight into why the PM biases occur.

Page 20, Line 519: Final dot is missing.

Page 21, Lines 570-571: The first reason had been marked with (i), so the second should be marked with (ii) for consistency. Also, this second reason is much less transparent here in the conclusions compared to (i), e.g. for a reader that just goes through the conclusions.

Page 22, Lines 582-583: These reasons are not mentioned in the main text, I think, and should be expanded a bit more either here or there.

---

## Referee Comment (RC2) · Anonymous Referee #2 · 31 May 2017

This paper describes a modeling system that nests the UK air quality forecast model (AQUM) with one way nesting into a regional composition-climate model covering Europe (RCCM) (50 km resolution) which itself is nested into global composition-climate model (GCCM) (140 km resolution). Evaluation is performed over a 5 year period in a regional climate type application. The paper claims to present an initial attempt to develop a single modelling framework, by introducing a greater degree of consistency in the modelling. Unfortunately, this does not include photolysis and chemistry. Overall I think the paper is interesting and deserves publication. I would, however, suggest to change the wording a little, to only claim consistency with respect to meteorology, since it appears that the physics parameterizations that are used in the different modeling systems are the same. This by itself is an important aspect of consistency, but this paper does not really provide proof for this. I did not grow up in an English-speaking country, so I leave any English corrections to reviewer 1. Overall the authors did a lot of work and summarize their work in this well written paper in a clear way. I therefore think this paper should be published with minor corrections. My main comments are:

Abstract, line3: You really only are more consistent with respect to the meteorological part of the modeling system. This should be stated.

Line 20/21: Where do you show that consistency between models is important? I believe you, but I do not see proof for this in your paper.

Introduction: You should find references for modeling systems that you cite: WRF-CMAQ, WRF-Chem, CESM, CESM-NCSU.

Section 2: A little table would be nice to get an easy look at what parameterizations and chemical modules are used. What atmospheric radiation scheme is used? You mention you have the capability to use radiative and microphysical feedbacks. Why did you switch them off? Is there any direct coupling of the convective parameterization to atmospheric radiation and photolysis? This could have a significant impact on Ozone evaluations (see also section 4.2.2). How complex is the aqueous phase chemistry that is being used (I am assuming you have some aqueous phase chemistry, since you allow for interaction with microphysics). For my understanding, in section 3 you mention that sea salt and dust emissions are computed interactively based on surface wind speed, but in section 2 you say that sea salt is diagnosed on ocean grid points. I am assuming that means sea salt is not advected or transported in any way? And there is no memory, so it is purely instantaneous and based only on wind speed? You also indicate that the missing proper treatment of sea salt could be a reason for poor performance of PM10 evaluation. Are there observations that can give you an idea on what the fraction of sea salt with respect to total PM10 is?

---

## Author Comment (AC1) · 26 Jul 2017

**Response to Referee 1 Comments**

Page 1, Line 2: I suggest appending "simultaneously" at the end of this sentence. Done

Page 1, Lines 14-15: "derived from a combination of modelling and observations" ->

be more specific, does this refer to reanalysis data, and if so, which?

Text modified to state: "effectively producing an analysis of annual mean surface pollutant concentrations."

Page 1, Line 19: Please subscript "10". Done

Page 1, Line 21: "consistency between nested models is also important" -> This looks like a key conclusion, so I suggest being more explicit about what is meant. In fact, after reading the manuscript I am not sure how the results presented in it lead to this conclusion. Can the authors please explain or remove?

Since one of the key differences between the RCCM and AQUM simulations for ozone arises from differences in the photolysis scheme, we believe that this study highlights the importance of aligning process modelling schemes as far as possible when comparing nested model runs. We have therefore modified the text to state: "This study highlights the point that the resolution of models is not the only factor in determining model performance - consistency between nested models is also important."

Page 2, Line 26: I suggest changing "constituents" to "pollutants", as otherwise it is a bit unconventional to mention ozone before CO2 and methane as important for climate. Done

Page 2, Lines 44-45: Not clear what is meant – are they more inhomogeneous than, e.g. cloud distributions?

We have clarified this statement with the modification:

"Air pollutant concentrations exhibit a high degree of spatial inhomogeneity compared to meteorological fields such as temperature and wind,…".

Page 2, Lines 46-47: Not sure why the second part of the sentence in the parenthesis

is relevant here. Removed parentheses

Page 5, Lines 140-143: Why has the 2D scheme been chosen instead of the more detailed approach? Is it due to the computational cost?

We have explained this point in the following text which has been added:

"We used this option in the GCCM and RCCM configurations mainly for two reasons. First, the extended tropospheric chemistry version of UKCA, UKCA-ExtTC, has been developed and extensively evaluated only with the 2D-photolysis model, and there was no time within the scope of this work for development and evaluation of UKCA-ExtTC coupled to the online photolysis model FastJ. Second, there is a non-negligible, albeit not prohibitively large, extra cost attached to using the online photolysis scheme FastJ over the 2D-photolysis scheme..."

Page 5, Lines 145-148: Perhaps this sentence would seem better placed earlier on, when the use of CLASSIC is first mentioned. Again, are UKCA aerosols not chosen because of computational cost?

We have explained this point and added the following text:

"Although UKCA does include an aerosol microphysics scheme, GLOMAP-mode (Mann et al., 2010), the simpler mass-based CLASSIC aerosol scheme (Jones et al., 2001; Bellouin et al., 2011) was used across the three MetUM configurations for the following reasons: (1) the UKCA-ExtTC chemistry scheme has historically only been coupled to the CLASSIC scheme and there was no time within the scope of the current study to couple it to GLOMAP-mode, (2) the operational air quality forecast model, AQUM, also uses CLASSIC as its aerosol scheme, and one of the aims of this work was to maximise the consistency in the treatment of both meteorology and composition across the three model domains, and (3) the computational cost of running both UKCA-ExtTC and GLOMAP-mode would have been prohibitively expensive.

Page 5, Line 157: Is 89 a higher or a lower number compared to what is used in the standard (non-ExtTC) UKCA?

We have added the text:

"This version of UKCA applies a more detailed gas-phase chemistry scheme that has a significantly larger number of chemical species – 89 chemical species in comparison to the 41 and 55 in the StdTrop and TropIsop chemistry schemes in O'Connor et al. (2014), respectively – and chemical reactions – 203 in UKCA-ExtTC in comparison to the 121 and 164 described in O'Connor et al. (2014)."

Page 5, Line 164: Please add space after "ozone". Also, I suggest adding "simulated" before "oxidant species". Done

Fig. 1: I suggest either removing the more coarsely distanced lines from the map, to avoid confusion, or to add tick marks next to them. I also suggest that the resolution of the different domains is mentioned again in the caption.

We have modified the figure. The caption has been modified to include the domain resolution as follows:

"Figure 1. Nested modelling domains. The rectangular boundary of the figure is an extract of the GCCM (resolution 140km) containing the RCCM domain (resolution 50km) plotted in blue and the AQUM domain (resolution 12km) in red.

Page 7, Line 214: Why were 20 years of spin-up needed?

We have explained in the text that a pre-existing simulation was used.

Page 8, Line 251: I suggest adding "in the focus area" at the end of this sentence. Done

Page 8, Lines 255-258: Does this imply that the calibration described just above will not mean much for AQUM performance? That is a bit confusing.

We have added the following text to clarify:

"The calibration was performed to ascertain that the best possible boundary conditions are applied to the innermost, national-scale domain. Mace Head station was chosen because it is representative of the large-scale background tropospheric ozone level in the study area and includes the impact of transcontinental influx of pollution from North America."

Page 11, Line 321: The fact that the simulation is for average conditions around year 2000 has been elusive throughout the Experimental Setup section. I suggest clarifying this at the very beginning of the section.

The following text has been added

"It should be pointed out here that the entire setup is intended to represent a decadal climatological mean state of near present day conditions encompasing the period from 1995 to 2005 and centred on the year 2000. This particularly applies to the meteorological drivers (sea surface temperature, SSTs, and sea ice cover) and the anthropogenic emissions of pollutants. The latter will be discussed in more detail in section 3.2"

Fig. 2: Why would some spatial structure existing in RCCM over central Britain disappear when moving to AQUM? Also, please place "-2" and "-1" in superscripts.

This is a feature of the contouring scale used in the figure. The caption has been amended with superscripts.

Page 13, Line 364: I am confused again – why 2001-2005 while earlier it was mentioned that the runs are designed for being representative of ~2000?

See above (under 'Page 11, Line 321')

Fig. 4: Please subscript "2", "3", "10" etc. in pollutant chemical formulas/abbreviations

(also in other parts of the text), and change "2p5" to "2.5".

Done

Page 14, Line 383: "sies" -> "sites". Done

Page 15, Line 386: Maybe I am wrong, but wouldn't the seasonality of emissions over the UK be similar across the different scales? Not sure why the global model in particular would have an issue with the seasonality of emissions.

We have modified the text to read:

"This is possibly due to the poor representation of the monthly variation of emissions over the UK in the global model which is then inherited by the higher resolution models. However, other processes such as boundary layer mixing or chemistry could equally contribute. Further work would be required to elucidate this clearly."

Page 15, Lines 404-405: So, the photolysis scheme seems responsible. The discussion below is useful, but maybe some further insight would be required here given this counter-intuitive behaviour, i.e. a more detailed model performing less well. At least some basic insight on whether key photolysis reactions for ozone ($NO_2$, $O_3$->$O_1D$) become faster or slower?

The text has been modified to read:

Although the photolysis rates relevant to $O_3$ , $j(NO_2)$ -> NO and $j(O_3)$ -> $O_1 D$, are known to be biased low in the off-line photolysis  scheme relative to both observations and on-line photolysis (Telford et al., 2013), the modelled $O_3$ bias in AQUM-h is reduced to +6.99 μgm−3 with the off-line scheme, which is marginally better than the RCCM.

Page 16, Lines 422-423: However, the more focused domain performs even worse, which should be mentioned. Done

Page 17, Lines 447-448: And AQUM performs somewhat better – worth mentioning. Done. Also, the fact that SO2 performs ok is some (admittedly not so solid) indication that sulphate may not be the main contributor to PM biases? May be worth considering, in order to provide a bit more insight into why the PM biases occur.

The magnitude of sulphate aerosol over Europe is not sufficient to play a major role in the observed biases. We have expanded the discussion of PM10 biases and the modified text now reads:

"Poor modelling performance for PM10 is a common feature of many global composition and regional air quality models (e.g. Colette et al., 2011; Im et al., 2015) and is often attributed to the unreliability of primary emissions of coarse component aerosol, both from anthropogenic and biogenic sources. In our simulations the lack of sea salt in modelled values over land points plays a significant role in this under-prediction. Putaud et al. (2010) estimate that over NW Europe sea salt contributes on average between 7% (kerbside sites) 12% (rural sites) of observed annual mean PM10 . In periods of strong winds and at sites close to the coast downwind of the sea values may be considerably higher. A related consequence of our lack of inclusion of sea salt is that our aerosol modelling does not include sodium nitrate and so this coarse component of secondary aerosol is also missing from our estimates."

Page 20, Line 519: Final dot is missing. Done

Page 21, Lines 570-571: The first reason had been marked with (i), so the second should be marked with (ii) for consistency. Also, this second reason is much less transparent here in the conclusions compared to (i), e.g. for a reader that just goes through the conclusions.

The text has been restructured and expanded. It now reads:

"The main reason for this is likely to be that in the UK, SO2 levels have fallen dramatically over the last 25 years and ambient concentrations are now generally the result of relatively low magnitude traffic emissions and much stronger emissions from a small number of industrial point sources. This results in an annually averaged mean concentration map over the UK which shows relatively little spatial structure, but with a small number of locations having much higher concentrations due to strong local emission sources (see the PCM 1 km plot in Fig. 6(b)). This low level background with little overall spatial structure limits the quantitative increases in spatial correlation with the PCM climatologies. Another The other reason maybe the impact of is related to the introduction and removal of strong point emissions sources affecting the comparison, as noted in section 4.3. Conclusions regarding the benefits of high resolution modelling for PM2.5 have been hampered in the present study due to the lack of observations over the study period. This pollutant consists of has both primary and secondary contributions and one might expect improvements in the modelling of the primary component by higher resolution modelling. However the magnitude of the improvement will depend on the relative sizes of primary and secondary components and it may well be that the contribution of the large secondary component masks any improvement in the representation of the primary component. For PM10 , model performance remains poor regardless of model resolution, with all three regional models (RCCM, AQUM and AQUM-h) failing to capture

the observed frequency distribution and having negative biases in the range -14.41 to -12.45 µgm$^{-3}$ . Based on the observed PM values analysed by Putaud et al. (2010), it is estimated that the lack of sea salt lowers the modelled PM10 annual mean values by around 12%. Additional important factors in the under-prediction of PM10 magnitudes include the absence of coarse component sodium nitrate aerosol, the poor representation of other coarse component primary emissions and poor modelling of the growth of aerosols to sizes in the coarse range."

Page 22, Lines 582-583: These reasons are not mentioned in the main text, I think, and should be expanded a bit more either here or there.

These points have now been discussed in the main text (Section 4.2.3) and the discussion in the conclusions section has been expanded to read:

"For PM10 , model performance remains poor regardless of model resolution, with all three regional models (RCCM, AQUM and AQUM-h) failing to capture the observed frequency distribution and having negative biases in the range -14.41 to -12.45 µgm$^{-3}$ . Based on the observed PM values analysed by Putaud et al. (2010), it is estimated that the lack of sea salt lowers the modelled PM10 annual mean values by around 12%. Additional important factors in the under-prediction of PM10 magnitudes include the absence of coarse component sodium nitrate aerosol, the poor representation of other coarse component primary emissions and poor modelling of the growth of aerosols to sizes in the coarse range."

---

## Author Comment (AC2) · 26 Jul 2017

**Response to Referee 2 Comments**

Abstract, line3: You really only are more consistent with respect to the meteorological part of the modeling system. This should be stated. Done

Line 20/21: Where do you show that consistency between models is important? I believe you, but I do not see proof for this in your paper.

This comment is similar to the point raised by Referee 1 and the response is also similar:

Since one of the key differences between the RCCM and AQUM simulations arises from differences in the photolysis scheme we believe that this study highlights the importance of aligning process modelling schemes as far as possible when comparing nested model runs. We have therefore modified the abstract text to state: "This study highlights the point that the resolution of models is not the only factor in determining model performance - consistency between nested models is also important."

Introduction: You should find references for modeling systems that you cite: WRF-CMAQ, WRF-Chem, CESM, CESM-NCSU. Done

Section 2: A little table would be nice to get an easy look at what parameterizations and chemical modules are used.

Table A4 has been added.

 What atmospheric radiation scheme is used?

This is stated in Table A4.

You mention you have the capability to use radiative and microphysical feedbacks. Why did you switch them off?

We have added the following explanation:

"The reasons for this were two-fold: (1) the primary goal of this study was on the simulation of air quality, and not on the impact of air quality on model dynamics, and (2) for statistical significance, much longer simulations are required when radiative and microphysical feedbacks are active (typically 20–30 model years as opposed to 5–7 years without these feedbacks)."

Is there any direct coupling of the convective parameterization to atmospheric radiation and photolysis? This could have a significant impact on Ozone evaluations (see also section 4.2.2).

There is no direct coupling but convection does influence photolysis through the radiation and cloud schemes in the online photolysis scheme used in AQUM. In the offline scheme used in the GCCM and RCCM the cloud is prescribed. A summary outline of the photolysis treatment in all three models is included in Table A4:

| | GCCM | RCCM | AQUM |
|---|---|---|---|
| Photolysis | Offline, 2-D model with prescribed cloud and aerosol | | Fast-J, using online cloud and aerosol |

How complex is the aqueous phase chemistry that is being used (I am assuming you have some aqueous phase chemistry, since you allow for interaction with microphysics).

This is stated in Table A4:

"oxidation of $SO_2$ by both $H_2O_2$ and $O_3$ to form dissolved $SO_4$

For my understanding, in section 3 you mention that sea salt and dust emissions are computed interactively based on surface wind speed, but in section 2 you say that sea salt is diagnosed on ocean grid points. I am assuming that means sea salt is not advected or transported in any way? And there is no memory, so it is purely instantaneous and based only on wind speed?

We have added the following text to Section 3.2 to clarify the treatment of sea salt and mineral dust:

"Mineral dust is a fully prognostic, advected species but, as mentioned in section 2.1, sea salt is not advected and makes no contribution to model aerosol concentrations over land."

You also indicate that the missing proper treatment of sea salt could be a reason for poor performance of PM10 evaluation. Are there observations that can give you an idea on what the fraction of sea salt with respect to total PM10 is?

We believe sea salt plays a role in the biases but is not the only factor. In section 4.2.3 we have added the following text which addresses this point:

"Poor modelling performance for PM10 is a common feature of many global composition and regional air quality models (e.g. Colette et al., 2011; Im et al., 2015) and is often attributed to the unreliability of primary emissions of coarse component aerosol, both from anthropogenic and biogenic sources. In our simulations the lack of sea salt in modelled values over land points plays a significant role in this under-prediction. Putaud et al. (2010) estimate that over NW Europe sea salt contributes on average between 7% (kerbside sites) 12% (rural sites) of observed annual mean PM10. In periods of strong winds and at sites close to the coast downwind of the sea values may be considerably higher. A related consequence of our lack of inclusion of sea salt is that our aerosol

modelling does not include sodium nitrate and so this coarse component of secondary aerosol is also missing from our estimates."